# Stable sequential dynamics in prefrontal cortex represents subjective estimation of time

Yiting Li[1,2†], Wenqu Yin[1†], Xin Wang[2], Jiawen Li[1,3], Shanglin Zhou[4], Chaolin Ma[1]*, Peng Yuan[2]*, Baoming Li[1,5]*

[1]Institute of Biomedical Innovation, Jiangxi Medical College, Nanchang University, Nanchang, China; [2]Department of Rehabilitation Medicine, Huashan Hospital, State Key Laboratory of Medical Neurobiology, Institute for Translational Brain Research, MOE Frontiers Center for Brain Science, MOE Innovative Center for New Drug Development of Immune Inflammatory Diseases, Fudan University, Shanghai, China; [3]The Second Clinical Medicine School of Nanchang University, Nanchang, China; [4]State Key Laboratory of Medical Neurobiology, Institute for Translational Brain Research, MOEFrontiers Center for Brain Science, Fudan University, Shanghai, China; [5]Institute of Brain Science and Department of Physiology, School of Basic Medical Science, Hangzhou Normal University, Hangzhou, China

*For correspondence:
chaolinma@ncu.edu.cn (CM);
pyuan@fudan.edu.cn (PY);
bmli@hznu.edu.cn (BL)

†These authors contributed equally to this work

Competing interest: The authors declare that no competing interests exist.

## eLife Assessment

This **useful** study reports how neuronal activity in the prefrontal cortex maps time intervals during which animals wait to reach a reward, with this mapping remaining consistent across days. While most claims are supported by **solid** evidence, the study could have benefitted from an improved experimental design to more clearly disambiguate correlations between neuronal patterns and not only time but also stereotypical behaviors and restraint from impulsive decisions. This study will be of particular interest to neuroscientists focused on decision-making and motor control.

**Abstract** Time estimation is an essential prerequisite underlying various cognitive functions. Previous studies identified 'sequential firing' and 'activity ramps' as the primary neuron activity patterns in the medial frontal cortex (mPFC) that could convey information regarding time. However, the relationship between these patterns and the timing behavior has not been fully understood. In this study, we utilized in vivo calcium imaging of mPFC in rats performing a timing task. We observed cells that showed selective activation at trial start, end, or during the timing interval. By aligning long-term time-lapse datasets, we discovered that sequential patterns of time coding were stable over weeks, while cells coding for trial start or end showed constant dynamism. Furthermore, with a novel behavior design that allowed the animal to determine individual trial interval, we were able to demonstrate that real-time adjustment in the sequence procession speed closely tracked the trial-to-trial interval variations. And errors in the rats' timing behavior can be primarily attributed to the premature ending of the time sequence. Together, our data suggest that sequential activity maybe a stable neural substrate that represents time under physiological conditions. Furthermore, our results imply the existence of a unique cell type in the mPFC that participates in the time-related sequences. Future characterization of this cell type could provide important insights in the neural mechanism of timing and related cognitive functions.

## Introduction

Time estimation is an essential function in the brain (*Buhusi and Meck, 2005*; *Tsao et al., 2022*; *Treisman, 1963*), since many crucial cognitive functions implicitly require a record of time, such as motor control (*Nobre and van Ede, 2018*; *Heideman et al., 2016*) or memory (*Cueva et al., 2020*; *Brody et al., 2003*). The neural substrate for time estimation in the brain has been studied for decades, and several modes of time coding emerged from these studies: (1) Time can be represented by the gradual change of activity levels in certain cells (ramping) *Balcı and Simen, 2016*; *Kim et al., 2013*; *Mendoza et al., 2018*; (2) Individual cells show selective activation at a specific time point (sequential) *Mello et al., 2015*; *Pastalkova et al., 2008*; *Gouvêa et al., 2015*; *Medina et al., 2000*; *Paton and Buonomano, 2018*; and (3) Population coding that showed complex patterns but stable dynamics in latent space (*Merchant et al., 2011*; *Shuler and Bear, 2006*; *Wang et al., 2018*). Interestingly, by training the animal to learn different lengths of waiting periods, several groups found that these 'time codes' exhibit scaling properties so that the number of cells for coding different lengths of time remains constant, while the activity can be compressed or stretched according to the duration of target time (*Mello et al., 2015*; *Shimbo et al., 2021*; *Egger et al., 2019*; *Xu et al., 2014*). While these findings provide strong evidence for a neural mechanism of time coding in the brain, true causal evidence at single-cell resolution remains beyond reach due to technical limitations. Although inhibiting certain brain regions (such as medial prefrontal cortex, mPFC, *Kim et al., 2009*) led to disruption in the performance of the timing task, it is difficult to attribute the effect specifically to the ramping or sequential activity patterns seen in those regions as other processes may be involved.

Lacking direct experimental evidence, one potential way of testing the causal involvement of 'time codes' in time estimation function is to examine their correlation at a finer resolution. However, a limitation in the experimental protocols of previous studies is that the animal learns a fixed time target, so that the scaling phenomenon is observed at the group level. Thus, there is a lack of evidence regarding whether the scaling happens rapidly at single-trial level to support time estimation.

In this study, we utilized a novel timing task in rat that allowed the animal to control the waiting period on its own will, so that we can observe the correlation of 'time code' scaling with behavioral waiting responses at individual trial resolution. We found robust sequential activities in the mPFC when the rats performed the task. To the best of our knowledge, we provide the first piece of evidence that the scaling effect is dynamic to account for the variation of waiting periods at individual trial level. And the rats were capable of subjectively adjusting the scaling factor for accurate time estimation. Intriguingly, we found that cells coding for the start or end of the waiting period undergo cross-session shifts, while the sequential time code remains stable over weeks. This surprising stability of the time code suggests an underlying mechanism that is different from apparently similar sequential activities seen in place coding or time coding in hippocampus CA1 (*Hainmueller and Bartos, 2018*; *Taxidis et al., 2020*), which showed dynamic shifts across sessions. Altogether, our study provides strong evidence for the online utilization of sequential time code in rats mPFC during a timing task. The unique rapid scaling and cell-identity stability of these sequential time code suggest a designated cell population for coding time in this region.

## Results

### Calcium imaging in mPFC during rats perform the timing task

We trained eight rats to perform a modified version of the timing task used in previous studies (*Xu et al., 2014*). In order to get the water rewards, water-deprived rats must poke their nose in a designated hole and maintain position (*Figure 1A*). The rat was free to start and end the nose poke at its own will, but only when the duration of the nose poke was above the minimal threshold would the rat receive a water reward. The rat did not receive additional punishment for nose poke duration below the minimal threshold, and can start a new trial when it was ready. Importantly, once the nose poke duration was over the minimum threshold, the amount of water reward was proportional to the total length of the nose poke duration. We trained the rat to perform this task in two phases. After the initial shaping for rats to associate nose poke with water rewards, the rats were trained on a short duration phase in which the minimal threshold was set to be 300ms. When the rats' correct performance reached above 70%, they advanced into a long duration phase and the threshold was set to be 1500ms (*Figure 1A*).

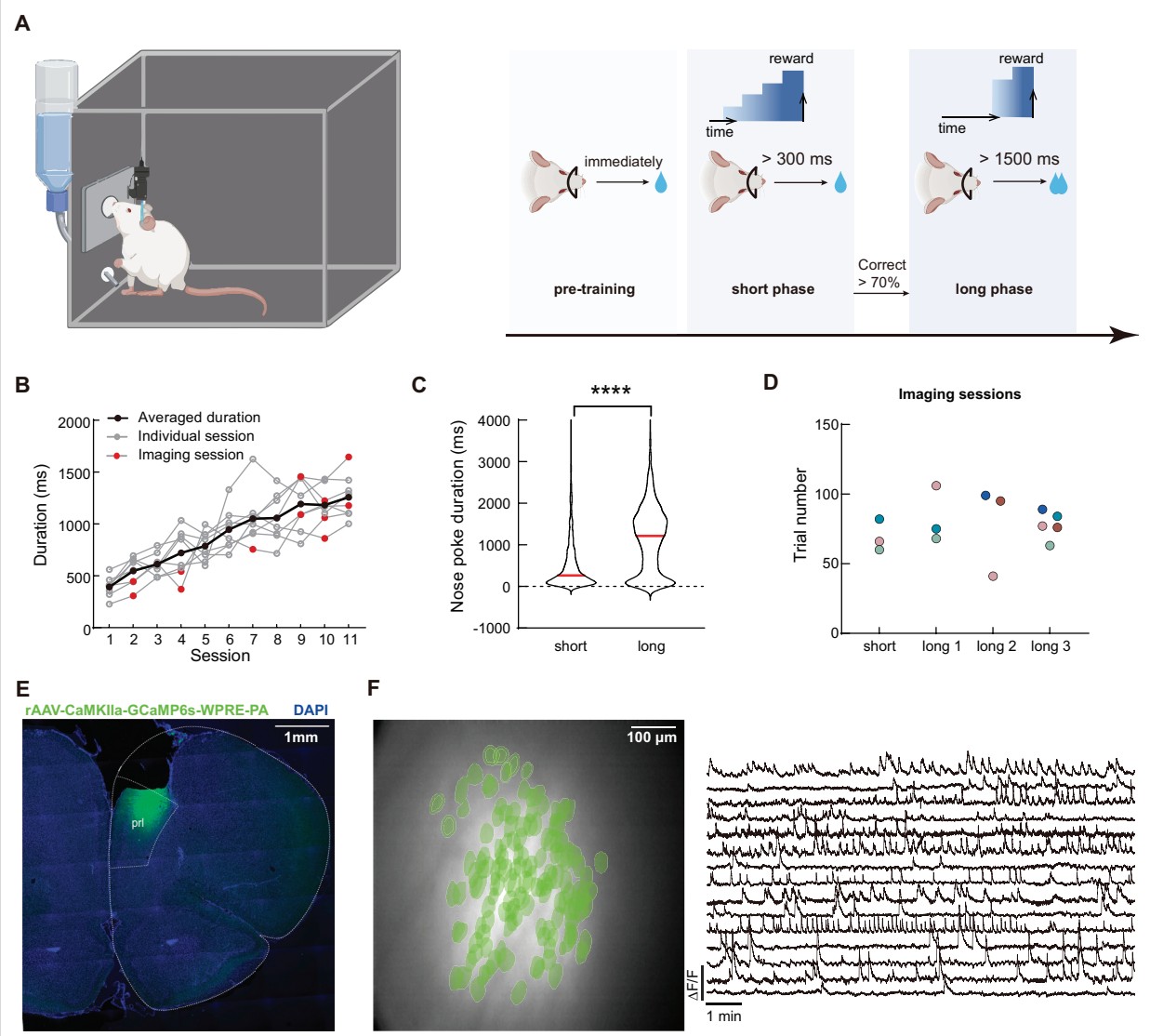

**Figure 1.** Calcium imaging during the timing task in rat mPFC. (**A**) Schematics of the timing tasks. Rats undergo behavior training in three phases. Pre-training phase: rat received water reward immediately after nose poking action. Short phase: rat had to maintain nose poking position for at least 300ms to receive reward. And long phase: rat had to maintain nose poking position for at least 1500ms to receive reward. Additional water reward would be provided for every additional 500ms of holding the nose poke. (**B**) Quantification of the rats' average nose poking durations in each session. Gray lines indicate each rat's performance, and the black line represents the averaged performance. Red dots indicate sessions with calcium imaging. (**C**) Violin plot for the nose poking duration in two phases of the task. Red lines indicate the median value in each group. Two-tailed t test, ****: p<0.0001, n=5 rats. (**D**) Sessions with calcium imaging data included in this study. The same color of dots indicated data from the same rat. (**E**) Representative histological section of mPFC from an imaged rat. The white dotted line indicates borders of brain structures. (**F**) Representative imaging field of view (left panel) with green masks of extracted cells' positions and example GCaMP fluorescent traces (right panel).

The online version of this article includes the following source data for figure 1:

**Source data 1.** Data for generating panels B, C and D.

Rats that underwent the training paradigm described above showed gradual increase in their nose poke durations (*Figure 1B*, *Figure 1—source data 1*). And the median durations in each phase were slightly above the respective threshold in that phase (*Figure 1C*, *Figure 1—source data 1*), indicating that the rats were capable of learning the different minimal threshold. Notably, there was a wide distribution of the nose poking durations in both phases of the task, which allowed us to study the trial-to-trial variation of the timing behavior.

We recorded neuronal activities from four different sessions from five rats (one from short phase and three from long phase, *Figure 1D*, *Figure 1—source data 1*). To obtain neuronal activity, we injected adeno-associated viruses expressing calcium sensor GCaMP6S in the medial prefrontal cortex (mPFC) of the rat and installed a 1.8 mm diameter GRIN lens after aspirating the cortical tissue on top (*Figure 1E*). Using a miniaturized microscope, we were able to record calcium activities from pyramidal neurons in the mPFC (*Figure 1F*), while the rats can freely move and perform the timing task. Our preparation yielded stable imaging data with good signal-to-noise ratio and cell counts (ranging from 50 to 150 cells per session, *Figure 2—figure supplement 1*, *Figure 2—figure supplement 1—source data 1*).

## Scaling of the mPFC time sequences at single-trial resolution

We started our analysis by examining the neural activities correlated with different features of the nose-poking behavior. To this end, we aligned calcium traces by nose-poking events for each cell, and calculated the Pearson's correlation between the averaged activity trace to the binarized event trace (start, during, and end). To determine the statistical significance of the correlation, we performed random shuffles of the calcium traces and repeated the calculations above, generating a distribution of correlation coefficients for each cell. If the actual correlation coefficient reached beyond 95% confidence interval of this distribution, we then assigned this cell to the group that codes for the specific feature of the nose-poking event. With this method, we identified neurons that showed selective activation at the beginning, the middle, or the end of the nose poke events (*Figure 2A*, *Figure 2—source data 1*), which we designated as 'start cell', 'duration cell' and 'end cell', respectively. A small portion of these coding cells (~10%) showed significant correlation between two features of the task ('start' and 'duration'; 'end' and 'duration'). In later analysis, we excluded these cells from the duration cell group. Notably, we observed gradual decays of GCaMP signal in start cells and sometimes gradual rise in end cells. However, we cannot definitively separate these 'ramping-like' activities from the potential artifact due to slow kinetics of calcium sensor (*Chen et al., 2013*). These three types of cells constituted more than half of the observed neurons (*Figure 2B*). By training a support-vector machine classifier using activities from the start cells and end cells, we were able to predict nose-poking events with above 90% accuracy, indicating that their activities were highly specific to the nose-poking events (*Figure 2C and D*, *Figure 2—source data 1*).

We then examined the activity patterns of the duration cells. We found that many duration cells showed activation during a certain proportion of the nose-poking events, and the timing of the activation seemed to be modulated by the total length of the events (*Figure 2E*, *Figure 2—figure supplement 2A*). When we normalized the actual time to the total length of each nose-poking event and transformed the calcium traces accordingly, we found that many duration cells showed selective activation at a fixed point in the normalized time scale (*Figure 2E*), consistent with the view that their actual activity was scaled by the total event duration (*Mello et al., 2015*; *Wang et al., 2018*). The ensemble activities from duration cells tiled across the normalized nose-poking time, showing a sequential activation pattern (*Figure 2F*). This activity sequence was stable across individual trials within a session, regardless of whether rat reached minimal reward threshold (*Figure 2G*, *Figure 2—figure supplement 1B-C*). These data suggest that the sequential activities of the duration cells may be representing nose-poking time.

To examine other possible interpretations of these data, we first measured the time interval between exiting nose poke to licking the water reward as indicators for the rat's motivation (*Figure 2—figure supplement 2*, *Figure 2—figure supplement 2—source data 1*). While nose-poking durations were correlated with this reward-seeking time, normalizing the duration cells' activities according to this motivation factor showed poor sequential patterns (*Figure 2F*), suggesting that the sequences were not representing the rat's motivation for water rewards. In addition, we measured the rat's head movements during the nose poke and found that the duration cells' activities were not modulated by these movements (*Figure 2—figure supplement 2*). Furthermore, we were able to train a Gaussian process regression model to predict the progress of each trial with high accuracy (*Figure 2H–J*, *Figure 2—source data 1*). While activities from start and end cells can also decode time, this might be due to slow calcium dynamics arising from either ramping activity or GCaMP kinetics as peak-extracted traces showed no decoding power (*Figure 2J*, *Figure 2—source data 1*). On the other hand, peak-extracted activities from duration cells maintained high decoding power, indicating a more reliable

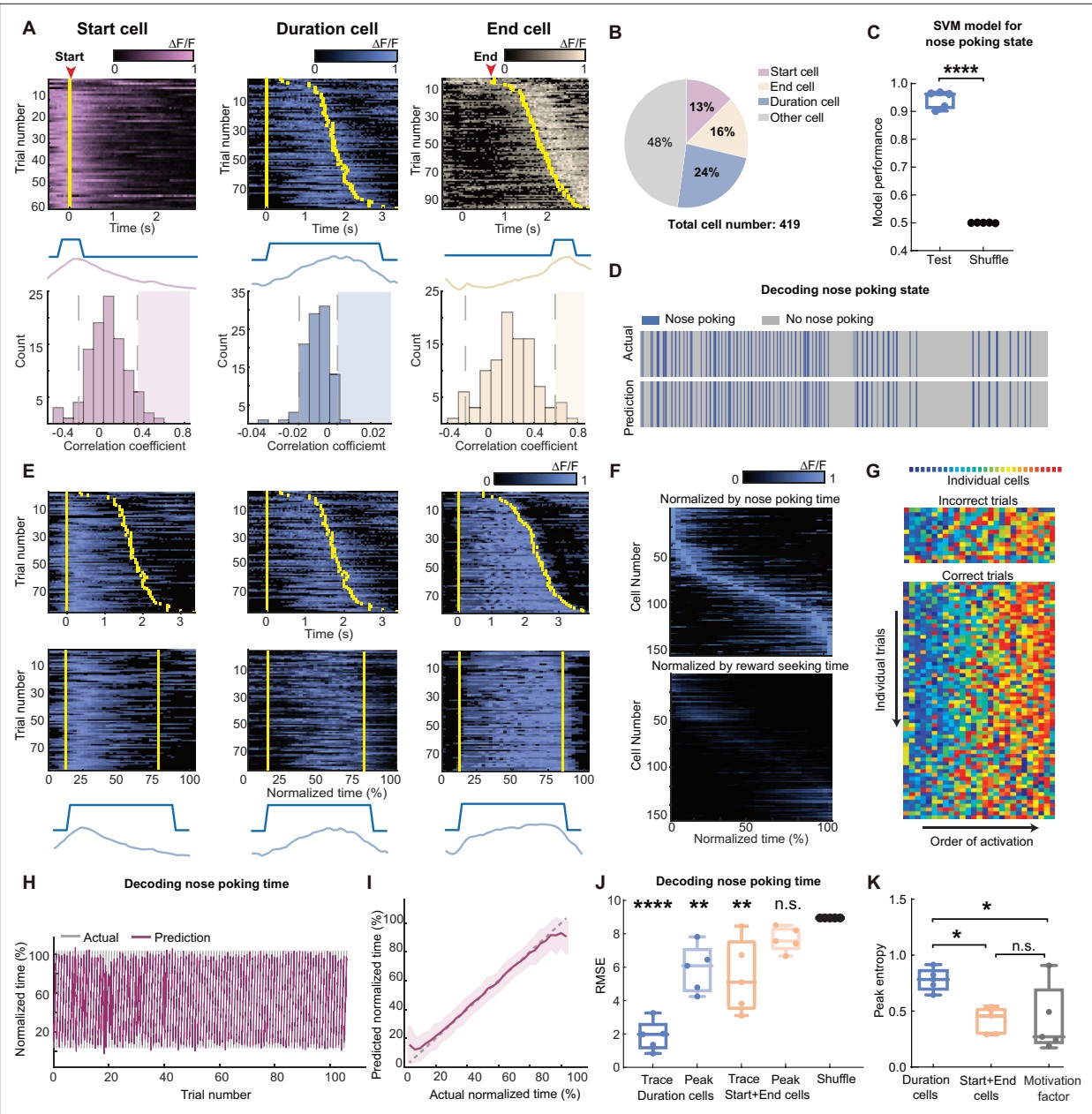

**Figure 2.** Scaling of the mPFC sequential activity by duration at individual trial resolution. (**A**) The definition and examples of start (pink), duration (blue), and end (beige) -coding cells. To define a cell's coding identity, we calculated the correlation between its activity and the specific event (middle diagrams), and compared to the null distribution generated from shuffled data from the same cell (bottom graphs). Statistical significance was determined if the actual correlation fell into the right 95th percentiles of the null distribution (shaded areas). Top panels show calcium activities from representative coding cells. Trials were sorted in ascending total durations. Yellow bars indicate the start and end of the nose poking event. (**B**) The proportion of cell coding types in the observed neurons from five rats in the long3 session. (**C**) Quantification of SVM classifiers in predicting the nose-poking state using neuron activities from the start and end cells, from the long3 session. Two-tailed t test, ****: p<0.0001, n=5 rats. (**D**) Representative actual nose poking events and predicted events using SVM models. (**E**) Three examples of duration cell with their raw traces (top panels) and traces in normalized time (middle panels). Yellow bars indicate the start and end of the nose poking event. Bottom panels show nose poking periods and averaged traces in normalized time scale. (**F**) Traces of averaged normalized activity of all recorded duration cells in five rats, sorted by the ascending peak position in the normalized time (upper panel) or the water reward-seeking time (lower panel). (**G**) Representative rainbow plot of the activity peak position ranks from duration cells in one session. Each color indicates an individual duration cell and the x-axis is sorted by the order of each cell's peak activity appearance on individual trials. (**H–I**) Representative results of predicted trial progression and actual trial progression by a GPR model fitted from duration cells activity in one session. Results are plotted in individual trials (**H**) and averaged trial (**I**). Results are represented as mean ± S.D. (**J**) Quantification of the GPR model performance. Each dot indicates averaged RMSE from one rat. Models were trained using duration cells or start and end cells activities, with raw calcium traces or extracted peaks. Data are represented as medium±S.D. One-way ANOVA test with Dunnett's multiple

*Figure 2 continued on next page*

*Figure 2 continued*

comparisons post-hoc test, ****: p<0.0001, **: p<0.01, n.s.: not significant, n=5 rats. (**K**) Quantification of the peak entropy on the trial-average calcium traces as a measure of sequentiality. The motivation factor group indicates duration cells' activities normalized with the water reward-seeking time. One-way ANOVA test with Dunnett's multiple comparisons post-hoc test, *: p<0.05, n.s.: not significant, n=5 rats.

The online version of this article includes the following source data and figure supplement(s) for figure 2:

**Source data 1.** Data for generating panels A, C, I, J and K.

**Figure supplement 1.** Additional examples of time-related sequential activities in mPFC.

**Figure supplement 1—source data 1.** Data for generating panel D.

**Figure supplement 2.** Analyses of rats' motivation and motions.

**Figure supplement 2—source data 1.** Data for generating panel F.

sequential activity pattern. Consistently, the trial-length normalized duration cell activity exhibited robust sequentiality as measured by peak entropy (*Zhou et al., 2020*; *Orhan and Ma, 2019*), which was not seen in activities from start or end cells, or when duration cells normalized with motivation factors (*Figure 2K*, *Figure 2—source data 1*). Together, these results indicated that distinct cell groups existed in the mPFC to code for different features of the nose-poking timing task. Duration cells showed selective activation towards specific timepoints in the normalized time scale, and their actual trial-by-trial activity was scaled by total event duration.

## mPFC sequential time code remains stable over weeks

Previous studies reported that sequential activity patterns in the hippocampus can represent temporally ordered events (*Rangel et al., 2014*) and timepoints (*Taxidis et al., 2020*). Interestingly, these sequences were highly dynamic, and the ordered pattern showed significant session-to-session variations (*Mankin et al., 2015*; *Mankin et al., 2012*). Therefore, we then examined whether the sequential time code we observed in the mPFC showed similar instability across different sessions. To achieve this, we aligned imaging data from multiple sessions based on matching unique blood vessel patterns in the field of view (*Figure 3—figure supplement 1*, *Figure 3—figure supplement 1—source data 1*). This allowed us to examine the coding properties in the aligned cells. We were able to see cells that maintained their coding features in different sessions (*Figure 3A–C*). And particularly for duration cells, we found that cells can represent a fixed timepoint in the duration across sessions, or shifted to another timepoint while still being a duration cell (*Figure 3C*, *Figure 3—figure supplement 1*).

We next formally quantified these dynamics. Below 30% of the start cells or end cells continued to maintain the same type of coding in the next session (*Figure 3D*). When we used the activities from the aligned start and end cells to classify the nose poking state, we found chance level performance for cross-session decoding despite of high within-session accuracy (*Figure 3E*, *Figure 3—source data 1*). These results indicated that cells code for nose poking start and end are highly dynamic and they did not form stable representation over days. In contrast, around 70% of the duration cells kept their identity in the next session. And within these stable duration cells, more than 70% of them coded for a fixed timepoint in the normalized time scale (*Figure 3F*). Using activities from aligned duration cells, we were able to make GPR models that show good cross-session decoding of the nose poking time (*Figure 3G*, *Figure 3—source data 1*), suggesting a stable code in the duration cells regarding the sequential activity pattern and time. We found largely the same results from three pairs of sessions: two sessions in the long phase with 2-day interval, two sessions in the long phase with 14-day interval and one short phase session and one long phase session with 16-day interval. Thus, the dynamism and stability we observed were not influenced by elapsed time or task structure, but may be a reflection of intrinsic properties of those cells. Given that previous studies in the hippocampus (*Mankin et al., 2012*) and cortex (*Tsao et al., 2018*) all showed unstable sequential pattern across days, to the best of our knowledge our data for the first time demonstrated a stable time code in the brain.

## Active scaling of the mPFC sequences represents subjective time estimation

Having found that mPFC showed time-associated stable sequential activities in the duration cells, we next investigated whether this activity could serve as a neural substrate for time estimation in rats performing our task. Previous studies showed that cooling of the mPFC in rats disrupted their

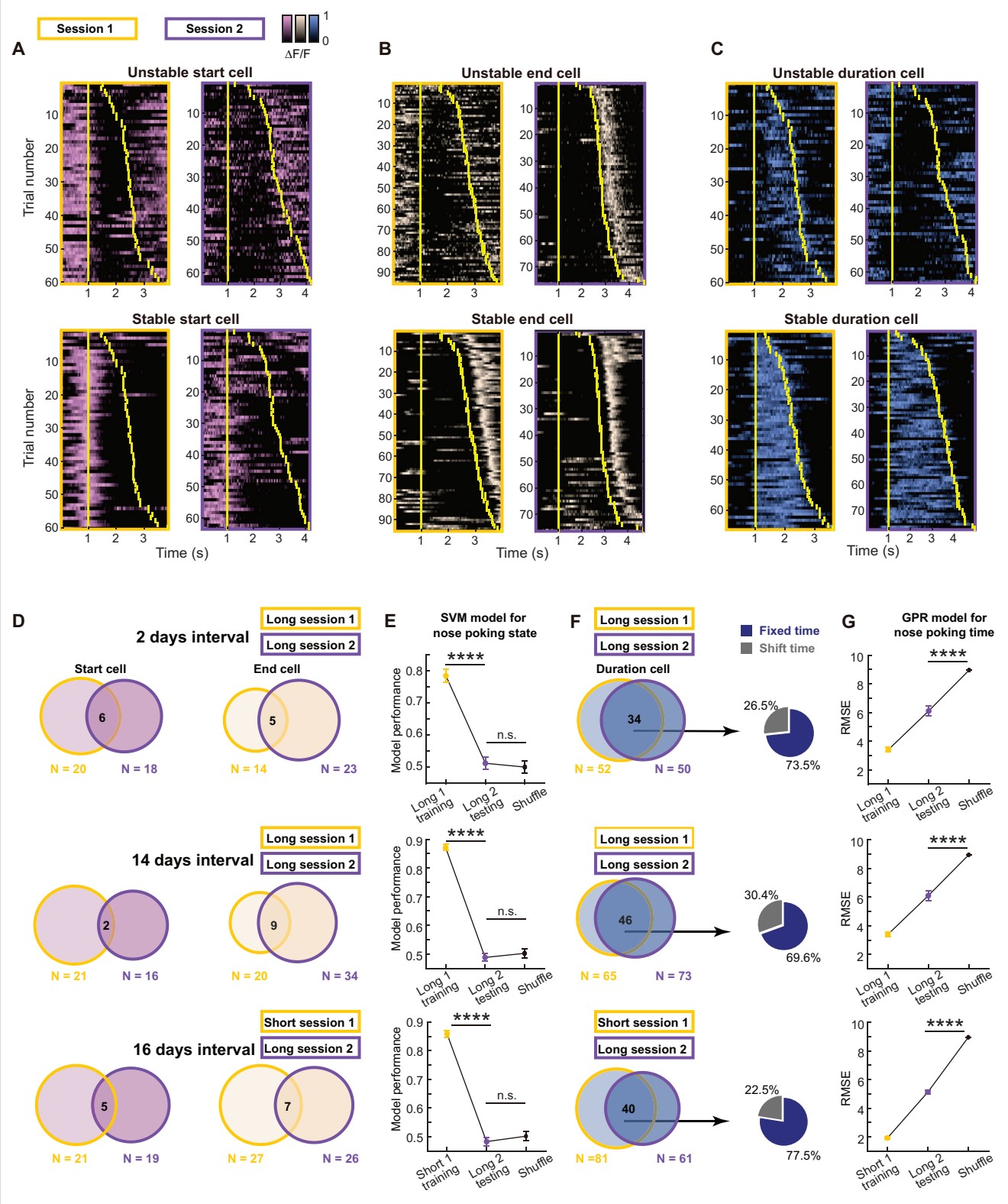

**Figure 3.** Long-term stable time coding by mPFC duration cell sequences. (**A–C**) Example traces of unstable or stable coding start cell pink, (**A**), end cell beige, (**B**) and duration cell blue, (**C**) in two sessions. Yellow bars indicate the onset and offset of the nose poking event. (**D**) Venn diagrams of dynamics in start cells and end cells across sessions with different intervals. (**E**) Quantification of within-session and cross-session performance of the SVM models trained by start cells and end cells from one session for classifying nose poking states. For each group, three rats' data were pooled together. Data are presented as mean ± S.D. One-way ANOVA test with Dunnett's multiple comparisons post-hoc test, ****: p<0.0001, ***: p<0.001, n.s.: not significant,

*Figure 3 continued on next page*

*Figure 3 continued*

n=50 random sampling. (**F**) Venn diagrams of dynamics in start cells and end cells across sessions with different intervals. And pie charts show shifted cell and fixed cell within the stable duration cells. (**G**) Quantification of within-session and cross-session performance of the GPR models trained by duration cells from one session for predicting normalized nose poking time. For each group, three rats' data were pooled together. Data are presented as mean ± S.D. Two-tailed t test, ****: p<0.0001, n=50 random sampling.

The online version of this article includes the following source data and figure supplement(s) for figure 3:

**Source data 1.** Data for generating panels E and G.

**Figure supplement 1.** Additional information related to cross-session alignment.

**Figure supplement 1—source data 1.** Data for generating panel A.

performance in tasks that require timing (*Xu et al., 2014*; *Kim et al., 2009*). However, in that study it was difficult to separate the influence of other cognitive functions affected by cooling the mPFC. Lacking the technology of single-neuron manipulation in freely moving rats, we speculated that further evidence on the necessity of mPFC sequential activities for time estimation could come from the analysis of the trials in that rats made errors. If the sequential time code we observed is causally linked with time estimation function, we should see reflection of behavior errors in the neural activity in some form. We hypothesized that there might be three types of coding errors in the mPFC: type I, absence of sequential activities that could be due to inattention or disengagement of the rats; type II, disordered sequences that may lead to errors in time estimation; and type III, scaling error with intact sequential code but wrong target time.

We then examined the above error types in detail. We performed a decomposition of the neural activity with principal component analysis in individual trials and found that almost all trials showed similar trajectories, indicating a lack of type I error described above (*Figure 4A*). Interestingly, we found that trajectories from correct trials appeared to be more expanded than those from the incorrect trials (*Figure 4B and C*), suggesting a difference in scaling between these two types of trials. We next utilized a partial least square regression model to examine the neural trajectory directions specifically aligned to trial durations (*Figure 4D*). Again, we observed that correct and incorrect trials in general follow the same direction during nose poke. In addition, we found that duration cells were more important for this trajectory pattern compared to other cells that we observed (*Figure 4E*). The constructed trajectories could explain ~52% of the variance in the normalized time, which was primarily contributed by the activities from duration cells (*Figure 4F*, *Figure 4—source data 1*). These results indicated a lack of type I or type II errors. Consistent with this view, when we examined the raw activities from duration cells in individual error trials, we found only a small fraction showed the absence of sequential activities (*Figure 4*, *Figure 4—figure supplement 1*), indicating that type III errors (error in activity scaling) are the primary source of coding error. Indeed, we can find some duration cells showed trial type-modulated activities (*Figure 4*, *Figure 4—figure supplement 1*, *Figure 4—figure supplement 1—source data 1*), suggesting that precise time estimation from these sequential activities may be affected.

To estimate the scaling errors associated with the type III errors, we calculated scaling factors of the sequential activities on individual trials. To this end, we first calculated a scaling factor with individual duration cells' activities by comparing the peak position of each cell's activity on individual trial to the expected position from the averaged template (*Figure 4G*). We found that correct trials showed significantly larger scaling factors compared to those from the incorrect trials (*Figure 4H and K*, *Figure 4—source data 1*), supporting the view that scaling errors are the major source for behavior errors. Furthermore, we computed a scaling factor following a previously reported method (*Xu et al., 2014*), by using population activities from the duration cells. We hypothesized that a singular multiplier could affect all duration cells in individual trials to prolong or compress the activity sequences, and the scaling factor for the trial should be the multiplier that minimizes the difference between the scaled activity and the averaged template (*Figure 4I*). The scaling factors derived this way again showed larger values in correct trials compared to incorrect ones (*Figure 4J nd L*, *Figure 4—source data 1*), further corroborating the importance of scaling errors.

Furthermore, we quantified the time estimation errors between the predicted time from duration cell neural activity and actual time using our previously established GPR models (*Figure 4M*). In general we did not find significant difference in prediction errors between correct and incorrect trials,

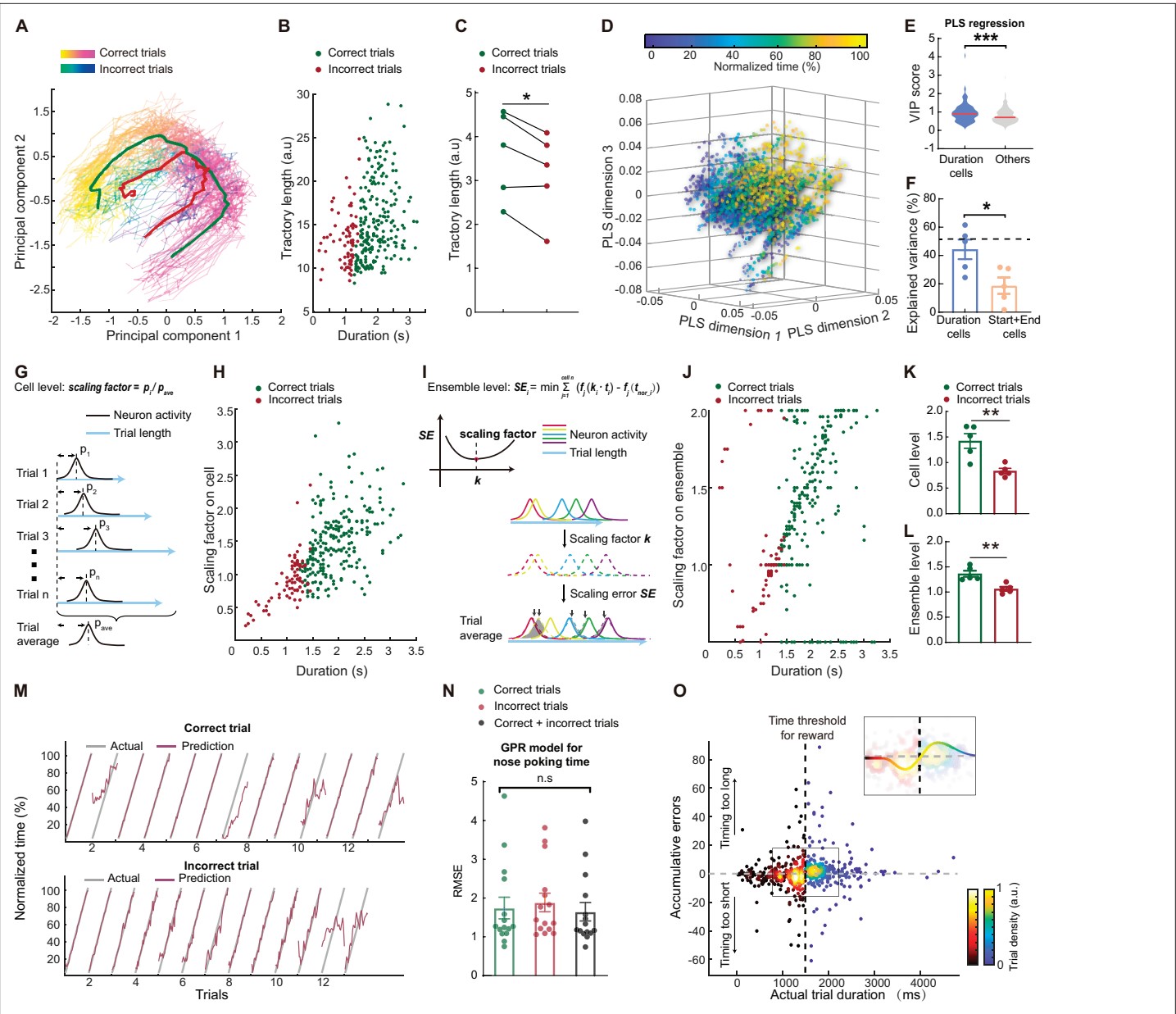

**Figure 4.** Behavioral errors in time estimation can be attributed to miscalculation in scaling the mPFC sequences. (**A**) Representative trajectories of neuronal activities from correct (spring colored) and incorrect (winter colored) trials. The red and green curves indicate averaged trajectories from incorrect and correct trials, respectively. (**B**) Scatter plot of the trajectories lengths from individual trials, plotted against trial durations. (**C**) Quantification of averaged correct and incorrect trajectory lengths. N=5 rats. Paired t test, *: p<0.05. (**D**) Scatter plots of the first 3 dimensions of a partial least-square (PLS) regression model between neuron activities and normalized trial time. (**E**) Violin plot of the variable importance for projection (VIP) scores derived from the PLS regression in **A**. Two-tailed t test, ***: p<0.0001. (**F**). Quantification of the explained variance in the normalized time by neuron activities from duration cells or start and end cells. Dashed line indicated explained variance from all cells. Data are represented as mean ± S.D. Two-tailed t test, *: p<0.05. n=5 rats. (**G**) Diagram of the scaling factor calculation process based on individual cell activities. (**H**) Scatter plot of the single-cell level scaling factor plotted against trial durations. (**I**) Diagram of the scaling factor calculation process based on population cell activities. (**J**) Scatter plot of the population level scaling factor plotted against trial durations. (**K and L**) Quantification of the averaged scaling factors on single-cell (**K**) or population (**L**) level. N=5 rats. Two-tailed t test, **: p<0.01. (**M**) Representative model prediction of normalized trial time for correct trials and incorrect trials. Each segment indicated on trial. The GPR model was trained on balanced correct and incorrect trials. (**N**) Quantification of GPR model performance for predicting different types of trials. In each group the model was trained on one type of trials and tested on the same trial type. Data are presented as mean ± S.D. One-way ANOVA test, n.s., not significant. (**O**) Scatter plots of cumulative prediction errors and actual trial duration. Correct and incorrect trials were colored differently but both by trial densities. There is a cluster of incorrect trials with below 0 cumulative errors and one for correct trials above 0. The insert shows the same data from the region in the box with a trend line color-coded with the trial densities.

*Figure 4 continued on next page*

*Figure 4 continued*

The online version of this article includes the following source data and figure supplement(s) for figure 4:

**Source data 1.** Data for generating panels F, H, J, K, L, N and O.

**Figure supplement 1.** Different types of incorrect types based on changes in mPFC time sequences.

**Figure supplement 1—source data 1.** Data for generating panel E.

**Figure supplement 2.** Graphic summary of the findings in current study.

and we were able to train a common model that showed accurate time estimation for both trial types (*Figure 4N*, *Figure 4—source data 1*), again indicating a lack of type I or type II error. To estimate the scaling errors, we calculated the cumulative errors between the model predicted time and actual trial time. When we plotted these data against actual individual trial durations, we found that interestingly, incorrect trials showed cumulative errors clustered below zero, and correct trials showed errors above zero (*Figure 4O*, *Figure 4—source data 1*), indicating that during the majority of the incorrect trials the rats timed too short from the expected sequences. Surprisingly, the duration time at which the polarity of the cumulative error reversed was 1500ms, exactly the minimal time threshold required for the rats to receive reward. This result strongly suggests that the rat was capable of perceiving the minimal time threshold and adjusting trial-by-trial scaling factor of the mPFC sequential activities during the task, and that errors in the online scaling accounted for most of the incorrect trials that we observed.

## Discussion

The neural basis of time representation is a fundamental question in neuroscience. As timing is closely intertwined with various cognitive functions, such as short-term memory, decision-making, etc., it is perhaps not surprising that neuron activities correlated with time were found throughout the brain. Previous studies established that sequential or ramping activity can be feasible neural code for time representation. Indeed, time-related sequential or ramping patterns of neuron activity have been reported in prefrontal cortex (*Meirhaeghe et al., 2021*; *Henke et al., 2021*), motor cortex (*Crowe et al., 2014*; *Merchant and Averbeck, 2017*), sensory cortex (*Liu et al., 2015*; *Chubykin et al., 2013*; *Namboodiri et al., 2015*), striatum (*Mello et al., 2015*; *Emmons et al., 2020*; *Toso et al., 2021*; *Kim et al., 2018*), thalamus (*Komura et al., 2001*), hippocampus (*Pastalkova et al., 2008*; *Itskov et al., 2011*; *MacDonald et al., 2011*), and entorhinal cortex (*Tsao et al., 2018*; *Diehl et al., 2019*). A fascinating feature of time coding is the capability of scaling, in which the same sequential or ramping patterns were compressed or stretched to represent different durations of time (*Tsao et al., 2022*; *Buonomano et al., 2023*). While these previous findings set forth substantial insights into the neural substrate of time in the brain, two major limitations exist in the field.

One aspect is the lack of direct causal evidence regarding the neuron activity and the perception of time. Resolving this would not be an easy task as the causal evidence between similar sequential patterns of place cell firing and spatial perception was not fully demonstrated either, although the phenomenon was described about 50 years ago. Recent advances in the technology indicated a possibility of single-cell optogenetic control for such experiments (*Robinson et al., 2020*), yet the low throughput and the difficulty for inhibitory control still greatly limit its application. While lacking direct causal evidence, results from our current study provides a finer correlation between sequential firing and timing function. As previous studies generally trained the animal to learn distinct categories of durations (*Xu et al., 2014*; *Bakhurin et al., 2017*), the scaling effect of sequential activity was usually described as a group effect. We discovered that scaling is correlated with durations at individual trial level, and that failure in reaching the minimal timing threshold for reward can be primarily attributed to scaling errors, in which the sequential pattern prematurely reached the end. Furthermore, the probability of high scaling error was disproportionally increased when the duration of the trial is close to the timing threshold for reward, suggesting that the animal provides trial-by-trial adjustment of the scaling factor that reflected its estimation of the timing threshold. The real-time scaling of sequential activity, and the enrichment of scaling errors around the timing threshold strongly suggest that the animal actively uses this mechanism, and thus the sequential activity we observed is likely to be the neural code for time estimation. We have also provided evidence excluding the effect of motivation

or motion. Future advances in single-cell manipulations may provide more definitive experimental evidence.

The second limitation in our understanding for the neural basis of time perception is the lack of biophysical mechanism for the emergence of sequential patterns and the scaling effect. In this regard, several theoretical models have been proposed (*Wang et al., 2018*; *Zhou et al., 2020*; *Hardy et al., 2018*; *Zhou and Buonomano, 2024*), yet testing these models in biological systems is currently not possible. Our data demonstrated a unique stability of the duration cells and their sequential activity patterns for coding time (*Figure 4—figure supplement 2*). Large proportions of duration cells' coding type and their sequential order remained unchanged over weeks, even when the target time threshold was changed. In contrast, within the same field of view, cells that code for the start or end phase of the nose poke showed dynamic changes, although we could not separate the start and end-coding cells from time-coding cells with ramping patterns and could not assess the long-term stability of time coding by ramping activities. Previous studies of sequential activity did not report such stability over time, and activities related to event trajectories are generally considered unstable (*Mankin et al., 2015*; *Mankin et al., 2012*). It is possible that this difference reflects different mechanisms underlying sequence emergence in different brain regions. This prolonged stability of duration cells suggests that in the mPFC, participation in the sequential pattern was unlikely a result of input-driven flexible coding, but a reflection of certain intrinsic properties. In other words, our data implied the existence of predetermined factors such as a genetic program for time coding neural population in the mPFC. Similarly, a recent study reported stable time sequences across different modalities in premotor cortex (*Betancourt et al., 2023*). A designated neural substrate for timing could be advantageous so that time coding can be a relatively independent module and can be inert to constant plastic changes happened in the brain. Future studies isolating molecular signatures of the time-coding population could provide a powerful tool in dissecting the neural mechanism of time representation in the brain.

# Materials and methods

## Subjects

Our study used Sprague Dawley rats. Equal number of male and female rats were used in the study. The environment temperature was controlled at 25°C with 12/12 light-dark cycle. All the rats were raised in single-housing after surgery with ad libitum access to food and water. During behavior training, water access was restricted according to an experimental procedure, in which the body weight loss was monitored and kept under 20%. Animal procedures were conducted in accordance with the Guiding Principles for the Care and Use of Laboratory Animals. All experiments were approved (protocol No.2017–0096) and monitored by the Ethical Committee of Animal Experiments at the Institute of Life Science, Nanchang University.

## Apparatus

Behavioral training and testing were conducted in a 40×20 × 30 cm plastic chamber. One side of the chamber had a semicircular hole (2.5 cm diameter) which was 13 cm above the floor and an infrared detector was placed to detect the nose poking, this hole was called as the operant hole. Under the operant hole is a reward hole where rats can get a certain volume of water, and water delivery is controlled by a solenoid valve. If the detector had detected a nose poking longer than the minimal time threshold from a rat, the solenoid valve will deliver a certain amount of water. We had written scripts to control those processes and record the timestamp of the nose poking.

## Virus injection

The rat was anesthetized with isoflurane and placed into a stereotactic apparatus. Rats were injected with carprofen solution (5 mg/kg) and dexamethasone solution (0.2 mg/kg) to minimize inflammation. We then shaved the head area and disinfected three times with alternating iodophors and alcohol. An opening on the skull was made with a dental drill. Then we unilaterally microinjected 500 nl of rAAV-CaMKIIa-GCaMP6S-WPRE-PA virus ($1.37e^{+12}$ vg/ml, BrainVTA) at 80 nl/min into the prelimbic cortex with the following stereotactic coordinates: 3.7 mm anterior to bregma, 0.5 mm lateral to the midline, and –3.2 mm ventral to the skull surface. After the virus injection, we waited 10 min to remove the

microelectrode from the prelimbic cortex, and then sutured the scalp and disinfected. The rat was placed back in the homecage after waking up from anesthesia.

## GRIN lens implantation

Our surgical method is based on the surgical procedure reported in mice (*Resendez et al., 2016*). Two weeks after the virus injection, the GRIN lens was implanted above the previous injection site. For the procedure, after disinfecting the head area, we removed the scalp with scissors and cleaned up the skull with hydrogen peroxide and saline, and then inserted three skull screws on the skull.The skull screws wereplaced in a way that would not block the lens. Then, we used saline to clean up the fragment, and drilled a hole with 1.9 mm diameter on top of the virus injection site. We slowly aspired brain tissue above the prelimbic cortex by connecting a syringe needle with the vacuum pump. The aspiration stopped at 2.5 mm below brain surface. We then inserteda GRIN lens(1.8 mm diameter, N.A: 0.54., Edmund) to the target position with the micromanipulators.The placement was checked using a miniscope(Labmaker) and adjusted to maximize the fluorescence signal.

We then removed excess liquid with the vacuum pump, and glued the GRIN lens with skull and skull screw using cyanoacrylate glue.After the glue was fully cured, we then applied black dental cement andcoveredthe top of the lens with Kwik-Sil (WPI) after the dental cement is dry. The ratwas placed back to their homecage and amoxicillin was added to their drinking water for 7 days.

After GRIN lens implantation, we waited 4–6 weeks to install baseplate. The rat was anesthetized with isoflurane again, and then we used forceps to remove the Kwik-Sil from the top of the lens, cleaned the lenswith wet lens paper. We then installed aminiscope with a micro-manipulator holder andfoundthebest view with clearvasculature and cells. At this position, we applied glue around the baseplate. After the glue was set, we continued to cover the larger area with dental cement. When the dental cement is dry, we removed the miniscope. Finally, we covered the plastic cap on the top of the baseplate and fixed it with a screw.

## The timing task

To motivate the rats to learn the task, their drinking water was limited during the training period, so that rats could only obtain water through task operation. Specifically, the rats need to probe their nose into the operant holeand stay longer than the minimal threshold time. After rats withdrew their nose from the hole, they can get reward from the reward hole.If the rats performed this action and acquired water reward, we called this as a correct trial. Conversely, an incorrect trial is when the rats started a nose pokebut did not reach designated time for reward.

The training was separated into three phases. In the pre-training phase, once rats produced the nose poking action, they could immediately acquire water reward at a small amount (60 µl). The rats moved on to the short phase after they mastered the nose poking action. In this phase, rats would acquire the reward only if they kept the action of nose poking for at least 300ms. Rats received 50 µl water with 300ms duration. The rats graduated from this phase when their performance reached above 70% correct. In the long phase, the nose poking duration must reach at least 1500ms. Rats received 100 µl water with 1500ms duration. In these two phases, with each additional 500ms nose poke time in a trial, the reward was increased by 60 µl.

## Calcium image recording

We did not anesthetize the rat for installing miniscope. Instead, we hand-held the rat to maintain its position and installed the miniscope and fixed it with screws. We monitored the calcium signal so that if the recording was not stable, the experiment would be interrupted. The calcium imaging of animals was recorded using a UCLA miniscope (V3), and the animal behavior was recorded using a webcam. Data acquisition software recorded the timestamps of the behavior camera and miniscope at the same time in order for subsequent alignment. Video streams were recorded at 20 frames per second. And excitation was set at ~4.5 mW. In our experiment, we used an MCX adapters to link the coaxial cable to ensure the animals can move freely and avoid the winding of cables.

## Extracting calcium activity

All analyses of the calcium imaging data were performed using custom software written in MATLAB. First, the raw imaging data was motion corrected using a previously published the NoRMCorre

algorithm (*Pnevmatikakis and Giovannucci, 2017*). The corrected movies have been processed by median filter and down-sampled prior to extract the calcium traces. Here, to accurately extract the calcium signals of individual cells, we applied a supervised and robust the EXTRACTalgorithm (*Inan et al., 2017*; *Inan et al., 2021*). Extracted units were checked manually to exclude artifacts.

## The identification of cell types

To identify cells that encode different periods of the nose-poking event, we calculated the Pearson'scorrelation coefficients between calcium traces and binarized event traces. In our task, we focused on three types of events, the start of nose pocking was defined as the 5 frames prior to the detection of the nose-poking event start. The end of nose probing was defined as the 5 frames after the rat began to withdraw from the operant hole. The duration of the nose poke was defined as the interrupted time of the infrared sensor in each trial. In order to calculate the statistics of the correlation coefficients, we generated a null hypothesis for each cell by shuffling the calcium traces with random delays for one hundred times, and calculated the resulted correlation coefficients. A cell was deemed to respond to certain events when the observed coefficient reached above the 95th percentiles of the corresponding coefficient distribution from the shuffled dataset. Those neurons responding specifically the three events were called respectively the start cells, the end cells and the duration cells.

## Calculation of peak entropy

PE (peak entropy) measures the entropy of peak times distribution across the population. Peak times are determined by analyzing the activity of the duration cells during normalized intervals for each trial. For a group of trials, the frequency of peak times is then calculated within M bins, where M represents the number of bins used to estimate the peak time distribution (30 frames for neural dynamics in normalized durations). Here, pj is the proportion of units peaking in time bin j relative to the total number of units. We did not further normalize the peak entropy with cell numbers since the numbers between groups were close to each other.

$$PE = \sum_{j=1}^{M} -p_j log\left(p_j\right)/log\left(M\right)$$

## The alignment of calcium imaging cross sessions

We performed cross-session alignment by manually select unique blood-vessel or neuron patterns as fixed control points, and then calculated the transformation between sessions by linear interpolation. We then applied the transformation to the centroid coordinates of each extracted cell to obtain the aligned cell positions in the target session. By calculating the pair-wise distances between the aligned centroids and that from the extracted cells in that session, we selected the minimal distance for each cell as the most likely candidate matches. And if this distance is below 5 pixels, which is about the radius of a cell, we recorded these two cells as a matched pair. In order to maximize the number of aligned cells, in all cross-session analysis, we used matched paired between two sessions.

## Support vector machine model for classification

We trained a linear kernel support vector machine (SVM) model using the activities of the start and end cells to predict the binarized nose poking state for each timepoint. In order to avoid biases caused by the imbalance of the samples, we chose random subsamples of recorded timepoints with equal number of nose-poking and non-nose-poking states. From this subsample, we then selected 80% of the timepoints for model training, and used the remaining 20% to evaluate the performance of the model. For performance measurement, we tabulated the differences between the model's predicted states from the actual states and calculated percentages of accurate predictions. In order to avoid variations introduced by sampling, we repeated our random sampling process for 100 times for each dataset and recorded the average performance. For generating a null hypothesis, the actual states were randomly shuffled during the training process, and the resulted model was then tested on real testing dataset. For testing cross-session performance, the within-session training and testing procedures were similar as described above, but using only cells that can be matched to the target session as model input. After training, the model was tested on the timepoints of the target session

using activities from the activities of the matched neurons, and accuracy calculation was the same as that described above.

## Gaussian process regression model

Neuron activities from each nose poking trial were isolated by taking the frames of each nose-poking bouts detected by the sensors and the amplitude of each cell's activity was normalized into a 0–1 scale within the trial. We then used linear interpolation algorithm (MATLAB 'interp1' function) to make each trial into the same number of 'normalized' timepoints. In our case we used 30 frames as the normalized trial length. Before model fitting, we selected subsamples of the same number of correct and incorrect trials, and used 80% of the trials for model training and saved 20% for testing. We then fitted a Gaussian process regression (GPR) model with a principal analysis pre-processing with a rational quadratic kernel function to describe the relationship between neuron activities and the progression of trial in normalized timepoints.

Rational Quadratic Kernel:

$$k\left(x_i, x_j|\theta\right) = \sigma_f^2 \left(1 + \frac{r^2}{2\alpha\sigma_l^2}\right)^{-\alpha}$$

$$r = \sqrt{\left(x_i - x_j\right)^T \left(x_i - x_j\right)}$$

where $\sigma$ is the characteristic length scale, $\alpha$ is a positive-valued scale-mixture parameter, $r$ is the Euclidean distance between $x_i$ and $x_j$.

We feed the testing set data to the fitted model to calculate the predicted normalized time. The performance was defined as the root mean square error (RMSE) between the actual normalized time and the predicted time across all tested trials. Similar to the SVM model, we repeated this process for 100 times to eliminate the sampling variations and shuffled the normalized time in the training set to generate a null hypothesis. It is notable that in this case, the null model output was generally around 15, which is at the middle of our normalized trial length. While this number gave no prediction power, the RMSE (~9) was lower than a random number set between 1 and 30 (~12).

## Partial least square regression

The VIP score was computed to authenticate the importance of neuronal ensembles on time estimation. VIP score is a measure used in PLS regression to identify the most important variables that predict a response variable. VIP score values range from 0 to 1, where higher values indicate greater importance of the variable in predicting the response. To calculate VIP score using the Partial Least Squares Regression method, several steps are involved. First, the dataset is divided into a training set and a test set. The training set is used to build the PLS regression model, while the test set is used to evaluate its performance. Next, the PLS regression model is trained using the training set. During this step, the model identifies the most relevant variables that explain the variance in the response variable. The VIP score for each variable is then calculated based on its contribution to explaining the variance in the response variable.

The VIP score is importance in predicting the response variable.

$$W0 = W_i / \sqrt{\sum_{i=1}^{n} \hat{W_i^2}}$$

$$sumSq = \sum_{i=1}^{n} \hat{XS_i^2} * \sum_{i=1}^{n} \hat{YL_i^2}$$

$$vipscore = \sqrt{n * \sum_{j=1}^{m} \left(sumSq_{ij} * \left(W0_{ij}^{\hat{2}}\right)\right) / \sum_{j=1}^{m} sumSq_{ij}}$$

$XS$ is an orthonormal matrix including $n$ time points by $m$ components. Each row of $XS$ corresponds to one time point, and each column corresponds to one component. $YL$ is an $n$-by-one matrix, where $n$ is the number of response variables and one is the number of PLS components. Each row of $YL$

contains coefficients that define a linear combination of PLS components approximating the original response variables. $W$ is $n$-by-$m$ components matrix of PLS weights.

## Principal components analysis (PCA) for duration cells

Weapplied the PCA function of matlab to determine the *score* of correct trials. The covariance matrixwas calculated for duration cell activities that included only the normalized time in the correct trials. Next, eigenvalue decomposition of the covariance matrix was performed to obtain eigenvalues and eigenvector matrices. The feature vectors were sorted according to the numerical magnitude of the feature values, and the top three feature vectors were selected as the principal components. Finally, the projection of the dataset on the selected principal components is calculated, that is, the score matrix is calculated. And the scores in incorrect trials were determined as the Principal component coefficient $coeff_{CT}$ and estimated means $\mu_{CT}$ from the data on the correct trials.

Incorrect trials:

$$score_{IT} = \left(activity_{IT} - \mu_{CT}\right) * coeff_{CT}$$

To quantify the trajectories lengths of duration cell activities, we calculated the length of the average trajectory on the correct trials and the average trajectory on the incorrect trials using the Euclidean distance. Here, the $n$ means that there are multiple time points in the average trajectory

$$trajectorylength = \sum_{i=1}^{n} \sqrt[2]{\left(x_i - x_{i+1}\right)^2 + \left(y_i - y_{i+1}\right)^2 + \left(z_i - z_{i+1}\right)^2}$$

## Scaling factor calculation

The scaling factor $k$ at the cellular level is calculated by dividing the location $p_i$ of the peak of activity of a single cell during each actual trial duration by the location $p_{ave}$ of the peak of average activity. The average activity here is taken as the number of trials with the highest frequency of nose poking time within a session, and the activity of individual neurons during these trials is averaged.

Cell level:

$$k = p_i / p_{ave}$$

To determine the optimal scaling factor $k_i$ at the ensemble level (where $k_i$ yields the minimum difference after scaling at ensemble level), we linearly compressed or stretched the binary activities of each duration cell within the same actual trial $t_i$ using various scaling factors ranging from 0.5 to 2. We calculated the standard error (SE) between the binary activities of each duration cell and their corresponding normalized time trial $\bar{t}_i$. The scaling factor $k_i$ corresponding to the $i$th trial that resulted in the minimum SE was selected as the best scaling factor. As the following equation:

Ensemble level:

$$SE_i = \min \sum_{j=1}^{cell\,n} \left(activity_j\left(k_i * t_i\right) - activity_j\left(\bar{t}_i\right)\right)$$

## Motion analysis

To explore the effect of motion on regulating neural activity during the timing task, we utilized the popular software DeepLabCut (*Mathis et al., 2018*; *Nath et al., 2019* for motion analysis based on transfer learning with deep neural networks. Using DeepLabCut, we successfully obtained information on the rat's head position. Initially, we created minimal training data from raw behavioral pictures where markers were manually placed on the rat's head. This labeled training data was then used with deep neural networks (DNNs) to track changes in the rat's head movement throughout the entire behavioral video. Specifically, movements during nose poking were extracted to analyze the duration of cell activity in relation to head movements.

## Statistics analysis

GraphPad Prism version 9.00 was used for statistical analyses. All data are presented as mean ± standard error (SE). Statistical significance was assessed by two-tailed and non-parametricStudent's t-tests. $p < 0.05$ was considered statistically significant. *$p < 0.05$; **$p < 0.01$; ***$p < 0.001$; ****$p < 0.0001$.

## Acknowledgements

This study was supported by Jiangxi Natural Science Foundation 20171ACB20002 to BML, and National Natural Science Foundation of China 31960171 to CLM. PY was additionally supported by Shanghai Pilot Program for Basic Research – FuDan University 21TQ1400100 (22TQ019), the Lingang Laboratory (grant no. LG-QS-202203–09) and National Natural Science Foundation of China (32371036).

## Additional information

### Funding

| Funder | Grant reference number | Author |
| --- | --- | --- |
| Jiangxi Natural science fundation | 20171ACB20002 | Baoming Li |
| National Natural Science Foundation of China | 31960171 | Chaolin Ma |
| Shanghai Pilot Program for Basic Research | 22TQ019 | Peng Yuan |
| Lingang Laboratory | LG-QS-202203-09 | Peng Yuan |
| National Natural Science Foundation of China | 32371036 | Peng Yuan |

The funders had no role in study design, data collection and interpretation, or the decision to submit the work for publication.

### Author contributions

Yiting Li, Data curation, Formal analysis, Investigation, Visualization, Methodology, Writing – original draft; Wenqu Yin, Data curation, Investigation; Xin Wang, Visualization; Jiawen Li, Data curation; Shanglin Zhou, Formal analysis; Chaolin Ma, Baoming Li, Conceptualization, Funding acquisition, Project administration, Writing – review and editing; Peng Yuan, Formal analysis, Funding acquisition, Writing – review and editing

### Author ORCIDs

Yiting Li (iD) https://orcid.org/0009-0003-5104-5516
Peng Yuan (iD) https://orcid.org/0000-0003-4051-4447
Baoming Li (iD) https://orcid.org/0009-0006-1554-9700

### Ethics

Animal procedures were conducted in accordance with the Guiding Principles for the Care and Use of Laboratory Animals. All experiments were approved and monitored by the Ethical Committee of Animal Experiments at the Institute of Life Science, Nanchang University. Every effort was made to minimize suffering.(protocol No.2017 0096).

Reviewer #1 (Public review): https://doi.org/10.7554/eLife.96603.3.sa1
Reviewer #2 (Public review): https://doi.org/10.7554/eLife.96603.3.sa2
Author response https://doi.org/10.7554/eLife.96603.3.sa3

## Additional files

### Supplementary files
• MDAR checklist

### Data availability
All data generated or analysed during this study are included in the manuscript. Source data file containing all the data used to generate the figures have been provided.

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
