## [Editor Report · eLife Assessment]

This **useful** study reports how neuronal activity in the prefrontal cortex maps time intervals during which animals wait to reach a reward, with this mapping remaining consistent across days. While most claims are supported by **solid** evidence, the study could have benefitted from an improved experimental design to more clearly disambiguate correlations between neuronal patterns and not only time but also stereotypical behaviors and restraint from impulsive decisions. This study will be of particular interest to neuroscientists focused on decision-making and motor control.

---

## [Referee Report · Reviewer #1 (Public review)]

Summary:

This paper investigates the neural population activity patterns of the medial frontal cortex in rats performing a nose poking timing task using in vivo calcium imaging. The results showed neurons that were active at the beginning and end of the nose poking and neurons that formed sequential patterns of activation that covaried with the timed interval during nose poking on a trial-by-trial basis. The former were not stable across sessions, while the latter tended to remain stable over weeks. The analysis of incorrect trials suggests the shorter non-rewarded intervals were due to errors in the scaling of the sequential pattern of activity.

Strengths:

This study measured stable signals using in vivo calcium imaging during experimental sessions that were separated by many days in animals performing a nose poking timing task. The correlation analysis on the activation profile to separate the cells in the three groups was effective and the functional dissociation between beginning and end, and duration cells was revealing. The analysis on the stability of decoding of both the nose poking state and poking time was very informative. Hence, this study dissected a neural population that formed sequential patterns of activation that encoded timed intervals.

Weaknesses:

It is not clear whether animals had enough simultaneously recorded cells to perform the analyzes of Figures 2-4. In fact, rat 3 had 18 responsive neurons which probably is not enough to get robust neural sequences for the trial-by-trial analysis and the correct and incorrect trial analysis. In addition, the analysis of behavioral errors could be improved. The analysis in Figure 4A could be replaced by a detailed analysis on the speed, and the geometry of neural population trajectories for correct and incorrect trials. In the case of Figure 4G is not clear why the density of errors formed two clusters instead of having a linear relation with the produce duration. I would be recommendable to compute the scaling factor on neuronal population trajectories and single cell activity or the computation of the center of mass to test the type III errors.

Due to the slow time resolution of calcium imaging, it is difficult to perform robust analysis on ramping activity. Therefore, I recommend downplaying the conclusion that: "Together, our data suggest that sequential activity might be a more relevant coding regime than the ramping activity in representing time under physiological conditions."

Comments on revisions:

The authors responded properly to my initial comments. However, I have three additional recommendations for the reviewed manuscript.

First, the paper urgently needs proofreading by a professional English editor. Second, Figure 4 must be divided in 2, it has too many panels and the resolution of the figure is low. Finally, please consider that what is called scaling factor in Figure 4G should be called something like neural sequence position index. A scaling factor in the timing literature implies that the pattern of activation of a cell contracts or expands according to the timed interval.

---

## [Referee Report · Reviewer #2 (Public review)]

In this manuscript, Li and collaborators set out to investigate the neuronal mechanisms underlying "subjective time estimation" in rats. For this purpose, they conducted calcium imaging in the prefrontal cortex of water-restricted rats that were required to perform an action (nose-poking) for a short duration to obtain drops of water. The authors provided evidence that animals progressively improved in performing their task. They subsequently analyzed the calcium imaging activity of neurons and identify start, duration, and stop cells associated with the nose poke. Specifically, they focused on duration cells and demonstrated that these cells served as a good proxy for timing on a trial-by-trial basis, scaling their pattern of actvity in accordance with changes in behavioral performance. In summary, as stated in the title, the authors claim to provide mechanistic insights into subjective time estimation in rats, a function they deem important for various cognitive conditions.

This study aligns with a wide range of studies in system neuroscience that presume that rodents solve timing tasks through an explicit internal estimation of duration, underpinned by neuronal representations of time. Within this framework, the authors performed complex and challenging experiments, along with advanced data analysis, which undoubtedly merits acknowledgement. However, the question of time perception is a challenging one, and caution should be exercised when applying abstract ideas derived from human cognition to animals. Studying so-called time perception in rats has significant shortcomings because, whether acknowledged or not, rats do not passively estimate time in their heads. They are constantly in motion. Moreover, rats do not perform the task for the sake of estimating time but to obtain their rewards are they water restricted. Their behavior will therefore reflect their motivation and urgency to obtain rewards. Unfortunately, it appears that the authors are not aware of these shortcomings. These alternative processes (motivation, sensorimotor dynamics) that occur during task performance are likely to influence neuronal activity. Consequently, my review will be rather critical. It is not however intended to be dismissive. I acknowledge that the authors may have been influenced by numerous published studies that already draw similar conclusions. Unfortunately, all the data presented in this study can be explained without invoking the concept of time estimation. Therefore, I hope the authors will find my comments constructive and understand that as scientists, we cannot ignore alternative interpretations, even if they conflict with our a priori philosophical stance (e.g., duration can be explicitly estimated by reading neuronal representation of time) and anthropomorphic assumptions (e.g., rats estimate time as humans do). While space is limited in a review, if the authors are interested, they can refer to a lengthy review I recently published on this topic, which demonstrates that my criticism is supported by a wide range of timing experiments across species (Robbe, 2023). In addition to this major conceptual issue that casts doubt on most of the conclusions of the study, there are also several major statistical issues.

Main Concerns

(1) The authors used a task in which rats must poke for a minimal amount of time (300 ms and then 1500 ms) to be able to obtain a drop of water delivered a few centimeters right below the nosepoke. They claim that their task is a time estimation task. However, they forget that they work with thirsty rats that are eager to get water sooner than later (there is a reason why they start by a short duration!). This task is mainly probing the animals ability to wait (that is impulse control) rather than time estimation per se. Second, the task does not require to estimate precise time because there appear to be no penalties when the nosepokes are too short or when they exceed. So it will be unclear if the variation in nosepoke reflects motivational changes rather than time estimation changes. The fact that this behavioral task is a poor assay for time estimation and rather reflects impulse control is shown by the tendency of animals to perform nose-pokes that are too short, the very slow improvement in their performance (Figure 1, with most of the mice making short responses), and the huge variability. Not only do the behavioral data not support the claim of the authors in terms of what the animals are actually doing (estimating time), but this also completely annihilates the interpretation of the Ca++ imaging data, which can be explained by motivational factors (changes in neuronal activity occurring while the animals nose poke may reflect a growing sens of urgency to check if water is available).

(2) A second issue is that the authors seem to assume that rats are perfectly immobile and perform like some kind of robots that would initiate nose pokes, maintain them, and remove them in a very discretized manner. However, in this kind of task, rats are constantly moving from the reward magazine to the nose poke. They also move while nose-poking (either their body or their mouth), and when they come out of the nose poke, they immediately move toward the reward spout. Thus, there is a continuous stream of movements, including fidgeting, that will covary with timing. Numerous studies have shown that sensorimotor dynamics influence neural activity, even in the prefrontal cortex. Therefore, the authors cannot rule out that what the records reflect are movements (and the scaling of movement) rather than underlying processes of time estimation (some kind of timer). Concretely, start cells could represent the ending of the movement going from the water spout to the nosepoke, and end cells could be neurons that initiate (if one can really isolate any initiation, which I doubt) the movement from the nosepoke to the water spout. Duration cells could reflect fidgeting or orofacial movements combined with an increasing urgency to leave the nose pokes.

(3) The statistics should be rethought for both the behavioral and neuronal data. They should be conducted separately for all the rats, as there is likely interindividual variability in the impulsivity of the animals.

(4) The fact that neuronal activity reflects an integration of movement and motivational factors rather than some abstract timing appears to be well compatible with the analysis conducted on the error trials (Figure 4), considering that the sensorimotor and motivational dynamics will rescale with the durations of the nose poke.

(5) The authors should mention upfront in the main text (result section) the temporal resolution allowed by their Ca+ probe and discuss whether it is fast enough in regard of behavioral dynamics occurring in the task.

Comments on the revised version

I have read the revised version of the manuscript and the rebuttal letter. My major concern was that the task used is not a time estimation task but primarily taps into impulse control and that animals are not immobile during the nose-poking epoch. I provided factual evidence for this (the animal's timing performance is poor and, on average, animals struggle to wait long enough), and I pointed to a review that discusses the results of many studies congruent with the importance of movement/motivation, not only in constraining the timing of reward-oriented actions during so-called time estimation tasks but also in powerfully modulating neuronal activity.

The authors' responses to my comments are puzzling and unconvincing. First, on the one hand, they acknowledge in their rebuttal letter the difficulty of demonstrating a neuronal representation of explicit internal estimation of time. Then, they seem to imply that this issue is beyond the scope of their study and focus in the rebuttal on whether the neuronal activity they report shows signs of being sensitive to movement and motivation, which they claim is independent of movement and motivation. This leads the authors to make no major changes in their manuscript. Their title, abstract, introduction, and discussion are largely unchanged and do not reflect the possibility that there are major confounding factors in so-called time estimation (rodents are not disembodied passive information processors) that may well explain some of the neuronal patterns. Evidently, the dismissive treatment by the authors is not satisfying. I will briefly restate my comments and reply to their responses and their new figure, which not only is unconvincing but raises new questions.

My comments were primarily focused on the behavioral task. The authors replied: "Studying the neural representation of any internal state may suffer from the same ambiguity [by ambiguity they meant that it is difficult to know if animals are explicitly estimating time]. With all due respect, however, we would like to limit our response to the scope of our results. According to the reviewer, two alternative interpretations of the task-related sequential activity exist." The authors imply that my comments are beyond the scope of their study. That is not true. My comments were targeted at the behavior of the animals, behavior they rely on to title their study: "Stable sequential dynamics in prefrontal cortex represents a subjective estimation of time." When I question whether the task and behavioral data presented are congruent with "subjective estimation of time," my comments are not beyond the scope of the study-they directly tackle the main point of the authors. Other researchers will read the title and abstract of this manuscript and conclude: "Here is a paper that provides evidence of a mechanism for animals estimating duration internally (because subjective time perception is assumed to be different from using clocks)." Still, there is a large body of literature showing that the behavior of animals in such tasks can be entirely explained without invoking subjective time perception and internal representation. How can the authors acknowledge that they can't be sure that mice are estimating time and then have such an affirmative title and abstract?

In my opinion, science is not just about forcing ideas (often reflecting philosophical preconceptions) on data and dismissing those who disagree. It is about discussing alternative possibilities fairly and being humble. In their revised version, I see no effort by the authors to investigate the importance of movement and motivation during their task or seriously engage with this idea. It's much easier to dismiss my comments as being beyond the scope of their results. According to the authors, it seems that movements and motivations play no role in the task. Still, the animals are water-restricted, and during the task, they will display decreased motivation (due to increased satiety), and their history of rewarded vs. non-rewarded trials will affect their behavior. This is one of the most robust effects seen across all behavioral studies. Moreover, the animals are constantly moving. Maybe the authors used a special breed of mice that behave like some kind of robots? I acknowledge that this is not easy to investigate, but if the authors did not use high-quality video recording or an experimental paradigm that allows disentangling motivational confounds, then they should refrain from using big words such as subjective time estimation and discuss alternative representations by acknowledging the studies that do find that movement and motivation are present during reward-based timing tasks and do in fact modulate neuronal activity, even in associative brain regions.

To sustain their claim that what they reported is movement-independent, the authors provided a supplementary figure in which they correlated neuronal activity and head movement tracked using DeepLabCut. I have to say that I was particularly surprised by this figure. First, in the original manuscript, there was absolutely no mention of video recording. Now it appears in the methods section, but the description is very short. There is no information on how these video recordings were made. The quality of the images provided in Figure S2 is far from reassuring. It is unclear whether the temporal and spatial resolution would be good enough to make meaningful correlations. Fast head/orofacial movements that occur during nose-poking can be on the order of 20 Hz. To be tracked, this would require at least a 40 Hz sampling rate. But no sampling information is provided. The authors should explain how they synchronized behavioral and neuronal data acquisition. Could the authors share behavioral videos of the 5 sessions shown in Figure S2 so we can judge the behavior of the animals, the quality of the video, and the possibility of making correlations?

Figure S2A-F: I am not sure why the authors correlated nose-poking duration (time estimation) and the duration between upper and lower nose-pokes (reward-oriented movement). It is not relevant to the issue I raised. Without any information about video acquisition frame rate, the y-axis legend (frame) is not very informative. Still, in Figure S2A-F, Rat 5 shows a clear increase in nose-poke duration, which is congruent with decreased impulsivity. Is the time coding different in this rat compared to other rats? There are some similar trends in other animals (Rat 1 and maybe Rat 3), but what is surprising is the huge variability (big downward deflections in the nose-poke duration). I would not be surprised if those deflections occurred after a long pause in activity. Could the authors plot trial time instead of trial number? How do the authors explain such a huge deflection if the animals are estimating time?

Regarding Figure S2H: I don't see how it addresses my concern. My concern is that some of the Ca activity recorded during nose-poking reflects head movements. The authors need to show if they can detect head movement during nose-poking. Aligning the Ca data relative to head movement should give the same result as when aligning the data relative to the time at which the animals pull out of the upper nose-poke.

Minor comments:

In their introduction, the authors wrote: "While these findings [correlates of time perception] provide strong evidence for a neural mechanism of time coding in the brain, true causal evidence at single-cell resolution remains beyond reach due to technical limitations. Although inhibiting certain brain regions (such as medial prefrontal cortex, mPFC,22) led to disruption in the performance of the timing task, it is difficult to attribute the effect specifically to the ramping or sequential activity patterns seen in those regions as other processes may be involved. Lacking direct experimental evidence, one potential way of testing the causal involvement of 'time codes' in time estimation function is to examine their correlation at a finer resolution."

This statement is inaccurate at two levels. First, very good causal evidence has been obtained on this topic (see Monteiro et al., 2023, Nature Neuroscience), and see my News & Views on the strengths and weaknesses of this paper. Second, their proposal is inaccurate. Looking at a finer correlation will still be a correlative approach, and the authors will not be able to disentangle motor/motivation confounds.

---

## [Author Response]

The following is the authors’ response to the original reviews.

**eLife Assessment**
This useful study reports how neuronal activity in the prefrontal cortex maps time intervals during which animals have to wait until reaching a reward and how this mapping is preserved across days. However, the evidence supporting the claims is incomplete as these sequential neuronal patterns do not necessarily represent time but instead may be correlated with stereotypical behavior and restraint from impulsive decision, which would require further controls (e.g. behavioral analysis) to clarify the main message. The study will be of interest to neuroscientists interested in decision making and motor control.

We thank the editors and reviewers for the constructive comments. In light of the questions mentioned by the reviewers, we have performed additional analyses in our revision, particularly aiming to address issues related to single-cell scalability, and effects of motivation and movement. We believe these additional data will greatly improve the rigor and clarity of our study. We are grateful for the review process of eLife.

**Public Reviews:**

**Reviewer #1 (Public Review):**
Summary:This paper investigates the neural population activity patterns of the medial frontal cortex in rats performing a nose poking timing task using in vivo calcium imaging. The results showed neurons that were active at the beginning and end of the nose poking and neurons that formed sequential patterns of activation that covaried with the timed interval during nose poking on a trial-by-trial basis. The former were not stable across sessions, while the latter tended to remain stable over weeks. The analysis on incorrect trials suggests the shorter non-rewarded intervals were due to errors in the scaling of the sequential pattern of activity.Strengths:This study measured stable signals using in vivo calcium imaging during experimental sessions that were separated by many days in animals performing a nose poking timing task. The correlation analysis on the activation profile to separate the cells in the three groups was effective and the functional dissociation between beginning and end, and duration cells was revealing. The analysis on the stability of decoding of both the nose poking state and poking time was very informative. Hence, this study dissected a neural population that formed sequential patterns of activation that encoded timed intervals.

We thank the reviewer for the positive comments.

Weaknesses:It is not clear whether animals had enough simultaneously recorded cells to perform the analyzes of Figures 2-4. In fact, rat 3 had 18 responsive neurons which probably is not enough to get robust neural sequences for the trial-by-trial analysis and the correct and incorrect trial analysis.

We thank the reviewer for the comment. Our imaging data generally yielded 50-150 cells in each session. The 18 neurons mentioned by the reviewer are from the duration cell category. We have now provided the number of imaged cells from each rat in the new Supplementary figure 1D. In addition, we have plotted the duration cells’ sequential activity of individual trials for each rat in new Supplementary figure 1B and 1C. These data demonstrate robust sequential activities from the duration cells.

In addition, the analysis of behavioral errors could be improved. The analysis in Figure 4A could be replaced by a detailed analysis on the speed, and the geometry of neural population trajectories for correct and incorrect trials.

We thank the reviewer for the suggestions. We have now performed analyses of the neural population trajectories as the reviewer suggested. We have calculated the neural population trajectories using the first two principal components of the neural activities during nose poke events. While both correct and incorrect trials show similar shapes of the trajectories, correct trials show more expanded paths, with longer lengths on average. These new results are now updated in Figure 4. Since type I or type II errors would likely generate trajectories not following the general direction which is different from our observations, these results are consistent with our conclusion that scaling errors contribute to the incorrect behavior timing in these rats.

In the case of Figure 4G is not clear why the density of errors formed two clusters instead of having a linear relation with the produce duration. I would be recommendable to compute the scaling factor on neuronal population trajectories and single cell activity or the computation of the center of mass to test the type III errors.

To clarify the original Figure 4G, the correct trials tended to show positive time estimation errors while the incorrect trials showed negative time estimation errors. We believe that the polarity switch between these two types suggests a possible use of this neural mechanism to time the action of the rats.

In addition, we have performed the analysis suggested by the reviewer in our revision. We calculated two types of scaling factors. On individual cell level, we computed the peak position of individual trials to the expected positions from averaged template. And on neural population level, we searched for a scaling multiplier to resample the calcium activity data and minimized the differences between scaled activity and the expected template. Using these two factors, we found that correct trials show significantly larger scaling compared to incorrect trials, consistent with our original interpretation that behavior errors are primarily correlated with scaling errors in the neural activities (type III error). These new results are now incorporated in Figure 4 and we have also updated the main text for the descriptions.

Due to the slow time resolution of calcium imaging, it is difficult to perform robust analysis on ramping activity. Therefore, I recommend downplaying the conclusion that: "Together, our data suggest that sequential activity might be a more relevant coding regime than the ramping activity in representing time under physiological conditions."

We agree with the reviewer, and have now modified this sentence in the abstract.

**Reviewer #2 (Public Review):**
In this manuscript, Li and collaborators set out to investigate the neuronal mechanisms underlying "subjective time estimation" in rats. For this purpose, they conducted calcium imaging in the prefrontal cortex of water-restricted rats that were required to perform an action (nosepoking) for a short duration to obtain drops of water. The authors provided evidence that animals progressively improved in performing their task. They subsequently analyzed the calcium imaging activity of neurons and identify start, duration, and stop cells associated with the nose poke. Specifically, they focused on duration cells and demonstrated that these cells served as a good proxy for timing on a trial-by-trial basis, scaling their pattern of actvity in accordance with changes in behavioral performance. In summary, as stated in the title, the authors claim to provide mechanistic insights into subjective time estimation in rats, a function they deem important for various cognitive conditions.This study aligns with a wide range of studies in system neuroscience that presume that rodents solve timing tasks through an explicit internal estimation of duration, underpinned by neuronal representations of time. Within this framework, the authors performed complex and challenging experiments, along with advanced data analysis, which undoubtedly merits acknowledgement. However, the question of time perception is a challenging one, and caution should be exercised when applying abstract ideas derived from human cognition to animals. Studying so-called time perception in rats has significant shortcomings because, whether acknowledged or not, rats do not passively estimate time in their heads. They are constantly in motion. Moreover, rats do not perform the task for the sake of estimating time but to obtain their rewards are they water restricted. Their behavior will therefore reflects their motivation and urgency to obtain rewards. Unfortunately, it appears that the authors are not aware of these shortcomings. These alternative processes (motivation, sensorimotor dynamics) that occur during task performance are likely to influence neuronal activity. Consequently, my review will be rather critical. It is not however intended to be dismissive. I acknowledge that the authors may have been influenced by numerous published studies that already draw similar conclusions. Unfortunately, all the data presented in this study can be explained without invoking the concept of time estimation. Therefore, I hope the authors will find my comments constructive and understand that as scientists, we cannot ignore alternative interpretations, even if they conflict with our a priori philosophical stance (e.g., duration can be explicitly estimated by reading neuronal representation of time) and anthropomorphic assumptions (e.g., rats estimate time as humans do). While space is limited in a review, if the authors are interested, they can refer to a lengthy review I recently published on this topic, which demonstrates that my criticism is supported by a wide range of timing experiments across species (Robbe, 2023). In addition to this major conceptual issue that cast doubt on most of the conclusions of the study, there are also several major statistical issues.Main Concerns(1) The authors used a task in which rats must poke for a minimal amount of time (300 ms and then 1500 ms) to be able to obtain a drop of water delivered a few centimeters right below the nosepoke. They claim that their task is a time estimation task. However, they forget that they work with thirsty rats that are eager to get water sooner than later (there is a reason why they start by a short duration!). This task is mainly probing the animals ability to wait (that is impulse control) rather than time estimation per se. Second, the task does not require to estimate precisely time because there appear to be no penalties when the nosepokes are too short or when they exceed. So it will be unclear if the variation in nosepoke reflects motivational changes rather than time estimation changes. The fact that this behavioral task is a poor assay for time estimation and rather reflects impulse control is shown by the tendency of animals to perform nose-pokes that are too short, the very slow improvement in their performance (Figure 1, with most of the mice making short responses), and the huge variability. Not only do the behavioral data not support the claim of the authors in terms of what the animals are actually doing (estimating time), but this also completely annhilates the interpretation of the Ca++ imaging data, which can be explained by motivational factors (changes in neuronal activity occurring while the animals nose poke may reflect a growing sens of urgency to check if water is available).

We would like to respond to the reviewer’s comments 1, 2 and 4 together, since they all focus on the same issue. We thank the reviewer for the very thoughtful comments and for sharing his detailed reasoning from a recently published review (Robbe, 2023). A lot of discussions go beyond the scope of this study, and we agree that whether there is an explicit representation of time (an internal clock) in the brain is a difficult question to be answer, particularly by using animal behaviors. In fact, even with fully conscious humans and elaborated task design, we think it is still questionable to clearly dissociate the neural substrate of “timing” from “motor”. In the end, it may as well be that as the reviewer cited from Bergson’sarticle, the experience of time cannot be measured.

Studying the neural representation of any internal state may suffer from the same ambiguity. With all due respect, however, we would like to limit our response to the scope of our results. According to the reviewer, two alternative interpretations of the task-related sequential activity exist: 1, duration cells may represent fidgeting or orofacial movements and 2, duration cells may represent motivation or motion plan of the rats. To test the first alternative interpretation, we have now performed a more comprehensive analysis of the behavior data at all the limbs and visible body parts of the experimental rats during nose poke and analyzed its periodicity among different trials. We found that the coding cells (including duration, start and end cells) activities were not modulated by these motions, arguing against this possibility. These data are now included in the new Supp. Figure 2, and we have added corresponding texts in the manuscript.

Regarding the second alternative interpretation, we think our data in the original Figure 4G argues against it. In this graph, we plotted the decoding error of time using the duration cells’ activity against the actual duration of the trials. If the sequential activity of durations cells only represents motivation, then the errors should be linearly modulated by trial durations. The unimodal distribution we observed (Figure 4G and see graph below for a re-plot without signs) suggests that the scaling factor of the sequential activity represents information related to time. And the fact that this unimodal distribution centered at the time threshold of the task provides strong evidence for the active use of scaling factor for time estimation.

In order to further test the relationship to motivation, we have measured the time interval between exiting nose poke to the start of licking water reward as an independent measurement of motivation for each trial. We found that this reward-seeking time was positively correlated with the trial durations, suggesting that the durations were correlated with motivation to some degree. And when we scaled the activities of the duration cells by this reward-seeking time, we found that the patterns of the sequential activities were largely diminished, and showed a significantly lower peak entropy compared to the same activities scaled by trial durations. The remaining sequential pattern may be due to the correlation between trial durations and motivation (Supp. Figure 2), and the sequential pattern reflects timing more prominently. These analyses provide further evidence that the sequential activities were not coding motivations. These data are included in Figure 2F, 2K and supp. Figure 3 in revised manuscript.

Regarding whether the scaling sequential activity we report represents behavioral timing or true time estimation, we did not have evidence on this point. However, a previous study has shown that PFC silencing led to disruption of the mouse’s timing behavior without affecting the execution of the task (PMID: 24367075), arguing against the behavior timing interpretation. The main surprising finding of our present study is that these duration cells are different from the start and end cells

in terms of their coding stability. Thus, future studies dissecting the anatomical microcircuit of these duration cells may provide further clues regarding whether they are connected with reward-related or motion-related brain regions. This may help partially resolve the “time” vs.

“motor” debate the reviewer mentioned.

(2) A second issue is that the authors seem to assume that rats are perfectly immobile and perform like some kind of robots that would initiate nose pokes, maintain them, and remove them in a very discretized manner. However, in this kind of task, rats are constantly moving from the reward magazine to the nose poke. They also move while nose-poking (either their body or their mouth), and when they come out of the nose poke, they immediately move toward the reward spout. Thus, there is a continuous stream of movements, including fidgeting, that will covary with timing. Numerous studies have shown that sensorimotor dynamics influence neural activity, even in the prefrontal cortex. Therefore, the authors cannot rule out that what the records reflect are movements (and the scaling of movement) rather than underlying processes of time estimation (some kind of timer). Concretely, start cells could represent the ending of the movement going from the water spout to the nosepoke, and end cells could be neurons that initiate (if one can really isolate any initiation, which I doubt) the movement from the nosepoke to the water spout. Duration cells could reflect fidgeting or orofacial movements combined with an increasing urgency to leave the nose pokes.(3) The statistics should be rethought for both the behavioral and neuronal data. They should be conducted separately for all the rats, as there is likely interindividual variability in the impulsivity of the animals.

We thank the reviewer for the comment, yet we are not quite sure what specifically was asked by the reviewer. It appears that the reviewer requires we conduct our analysis using each rat individually. In our revised manuscript, we have conducted and reported analyses with individual rat in the original Figure 1C, Figure 2C, G, K, Figure 4F.

(4) The fact that neuronal activity reflects an integration of movement and motivational factors rather than some abstract timing appears to be well compatible with the analysis conducted on the error trials (Figure 4), considering that the sensorimotor and motivational dynamics will rescale with the durations of the nose poke.(5) The authors should mention upfront in the main text (result section) the temporal resolution allowed by their Ca+ probe and discuss whether it is fast enough in regard of behavioral dynamics occurring in the task.

We thank the reviewer for the suggestion. We have originally mentioned the caveat of calcium imaging in the interpretation of our results. We have now incorporated more texts for this purpose during our revision. In terms of behavioral dynamics (start and end of nose poke in this case), we think calcium imaging could provide sufficient kinetics. However, the more refined dynamics related to the reproducibility of the sequential activity or the precise representation of individual cells on the scaled duration may be benefited from improved time resolution.

**Recommendations for the authors:**

**Reviewer #1 (Recommendations For The Authors):**
(1) Please refer explicitly to the three types of cells in the abstract.

We have now modified the abstract as suggested during revision.

(2) Please refer to the work of Betancourt et al., 2023 Cell Reports, where a trial-by-trail analysis on the correlation between neural trajectory dynamics in MPC and timing behavior is reported. In that same paper the stability of neural sequences across task parameters is reported.

We have now cited and discussed the study in the discussion section of the revised manuscript.

(3) Please state the number of studied animals at the beginning of the results section.

We have now provided this information as requested. The numbers of rats are also plotted in Figure 1D for each analysis.

(4) Why do the middle and right panels of Figure 2E show duration cells.

Figure 2E was intended to show examples of duration cells’ activity. We included different examples of cells that peak at different points in the scaled duration. We believe these multiple examples would give the readers a straight forward impression of these cells’ activity patterns.

(5) Which behavioral sessions of Figure 1B were analyzed further.

We have now labeled the analyzed sessions in Figure 1B with red color in the revised manuscript.

(6) In Figure 3A-C please increase the time before the beginning of the trial in order to visualize properly the activation patterns of the start cells.

We thank the reviewer for the suggestion and have now modified the figure accordingly in the revised manuscript.

(7) Please state what could be the behavioral and functional effect of the ablation of the cortical tissue on top of mPFC.

We thank the reviewer for the question. In our experience, mice with lens implanted in the mPFC did not show observable difference with mice without surgery in the acquisition of the task and the distribution of the nose-poke durations. In our dataset, rats with the lens implantation showed similar nose-poking behavior as those without lens implantation (Figure 1B). Thus, it seems that the effect of ablation, if any, was quite limited, in the scope of our task.